# Spatial price transmission in the onion markets of Bangladesh: An application of NARDL approach

**Farhana Arefeen Mila**[1⊕], **Ashrafun Nahar**[1⊕]*, **Md. Emran Hossain**[2], **Md. Ruhul Amin**[1]

1 Department of Agribusiness, Bangabandhu Sheikh Mujibur Rahman Agricultural University, Gazipur, Bangladesh, 2 Department of Agricultural Finance and Banking, Bangladesh Agricultural University, Mymensingh, Bangladesh

⊕ These authors contributed equally to this work.
* anlaboni@bsmrau.edu.bd

**Data Availability Statement:** All relevant data are within the Supporting Information file.

**Funding:** The study was funded by the University Grants Commission, Bangladesh (S.R. No: 6(79)/B. Mo.ko/B.O.Pro/Crop-16/2018/7032) under the

## Abstract

As the price of local onions is greatly impacted by the price of imported onions at these two levels of the supply chain, the goal of this study was to ascertain whether onion prices in Bangladesh are transmitted asymmetrically at the wholesale and retail levels. To analyze asymmetry, the study employed the nonlinear autoregressive distributed lag (NARDL) model in the short and long run using monthly time series data from January 2006 to December 2020. The NARDL model captures the effects of both positive and negative shocks in the short as well as in the long run. The empirical results from the NARDL indicate that the wholesale price of local onion shares a short-run relationship with the wholesale price of imported onion while the local retail price of onion shares a long-run relationship with the imported retail price of onion. In addition, the short-run impact of local wholesale and imported wholesale prices is asymmetric. Long-run evidence supports the existence of an asymmetric effect between the local and imported retail onion prices. Using the Pairwise Granger causality test, we examined the causal relationships between wholesale and retail prices. The direction of the casual relationship indicates that the wholesale and retail prices of imported onions lead to the wholesale and retail prices of local onions. A clear understanding of the onion market, how prices move between market actors, and its role in determining market price interaction could be gained by analyzing the asymmetric relationship between the local and imported onion prices. As a result, significant policy recommendations could be made to control the onion price in Bangladesh.

## Introduction

Despite being a small, and agrarian nation, Bangladesh produces a lot of grain, pulse, cash crops, spices, and condiments. Among them, onion is the most important seasonal minor crop, since it plays a vital role in human nutrition, additive for the human consumption, generates rural and national income, and provides employment [1]. In Bangladesh, onion

entitled project "Analysis of Supply Chain and Price Transmission Dynamics in the Onion Market of Bangladesh". The funders played a major role in the study design, project report and decision to publish

**Competing interests:** The authors have declared that no competing interests exist.

producers get a higher net return per hectare than other species which encourages them to sell 86% of their total output [2]. Although domestic onions are profitable, from 2010–11 to 2019–20, the yearly production increase remained at 1.85% [3]. Just a few years back, Bangladesh was within the top ten onion-producing countries [4] but now it relies mostly on imports from neighboring countries [5]. According to the Ministry of Agriculture, the yearly demand for onions is 2.3–2.5 million metric tons, and the country imports between 0.9 and 1.1 million tonnes to make up for the shortfall [6]. It indicates that we have to import around 40% of onion from India and other neighboring countries to meet up the annual demand. As a result, the price of onions imported into Bangladesh significantly influences and stabilizes the price of onions in the domestic market. Different study on onion supply in Bangladesh indicates multiple intermediaries existence in the system [7–9]. There is a significant gap between producer price and consumer price of onion [10, 11]. This change in price usually occur frequently and sudden price rise and fall in domestic commodity market leads to misconception and undesired perception among the population which needs immediate attention, and the first step is to verify through understanding how the price formation and transmission of the different stage of supply chain take place in a market [12].

A long-standing and important issue in agricultural product marketing is price transmission along the channel. In many agricultural markets, the transmission of farm price rises to retail has been found to be larger and faster than the transmission of farm price drops. The food retailing, wholesaling, and processing sectors have become increasingly concentrated in recent years, resulting in food marketers and processors to have more substantial implications for farmers and consumers than before [13]. The link between farm and retail pricing offers perceptions into the effectiveness of marketing and the wellbeing of consumers and farmers. For the sake of producer and consumer welfare, agricultural economists have concentrated on the farm-to-retail or wholesale to retails price transmission process. Price transmission identification of different commodities around the world is a widely popular practice because of its importance. Studies on price transmission can aid policymakers in numerous ways. For instance, nations that liberalize their domestic markets must understand how global price signals are transmitted to their home markets. Understanding effective price transmission, which leads to the integration and efficiency of spatially separated markets, is vital for maintaining a distributional equilibrium between food-deficit and surplus regions in developing nations. The role of profit-seeking individuals and firms must also be evaluated in this context [14]. In addition, research on price transmission is required to understand the influence of policy changes on the operation of agricultural markets. Thus, this research is crucial for comprehending the ability of domestic agricultural markets in developing nations to respond to changes in the international prices of agricultural commodities.

In past years, not only in developing countries like Bangladesh, India, Pakistan but also in developed countries (i.e., USA, UK, Japan) the product price transmission from one market to another market level was also identified. Market power is also related to imbalance of market price transmission in developing countries like South Africa [11]. In USA, pork sector, beef industry sector, supermarket fluid milk sector, fresh fruit sector were identified to have asymmetric price transmission both in term of price and speed of product flow [13, 15–18]. Similarly, in many other countries of the world, the price transmission assessment was done on different products. Price asymmetry is also common in European and French fish market, Chinese pork and pig market, Thailand dairy market, Turkish milk and meat market, Brazilian commercial rice market [19–23]. In onion market, several studies were found in India depicted an active integration among the wholesale and retail markets either bi-directional or unidirectional [24–26] along with positive and negative price distortion between these vertical markets [26]. Theoretically, farm supply, consumer demand, and buyer-seller power of controlling

market causes asymmetry in the price transmission from farm market to retail market [12]. In Europe, agriculture market goes through positive price transmission asymmetry (PTA) which indicates increased market power at retail level over processor level [27].

In Bangladesh, asymmetric price transmission due to consumer belief is confirmed through vertical supply chain in deregulated rice market [28]. A short-run symmetry and long run symmetry-asymmetry mix was identified in the long run for seafood value chain in Bangladesh [29]. In wheat and flour market, short run price asymmetry was found while in long run the adjustment is different [30]. Welfare impact of asymmetric rice price transmission in Bangladesh market was also assessed [31]. Similar price transmission in both short and long run is also observed for edible oil market [32]. Being one of the important agricultural item, onion price fluctuation is most frequent [33–36] and in short run this price change is rapid. Moreover, the price of local onion at wholesale and retail level are greatly influenced by the imported onion price. For this reason, the government works to increase local onion production; nevertheless, imports of onions are controlled and handled by a small group of businesspeople, which results in a more centralized industry and an increase in domestic prices under the sway of this syndicate. Additionally, there is a significant price differential between wholesale and retail, which emphasizes the need for more research on price transmissions. Considering this, this study aims to analyze the presence or absence of asymmetric price transmission (APT) among the two stages (wholesale and retail) prices for both local and imported onion in the short and long run. The specific objective of this study is to identify the extent to which changes in the wholesale and retail price of imported onions are transmitted to wholesale and retail price of local onions. For example, if the wholesale price of imported onions rises, it is expected that the wholesale price of local onions will also increase. However, the extent of this price increase and the speed of transmission may differ from that of local onions. Thus, the non-linear autoregressive distributed lag (NARDL) model is used in this study to account for the price transmission of domestic and imported onions at various supply chain levels. The short-run dynamic process and the long-run asymmetric cointegration are combined in the NARDL model, which has the benefit of preventing biased outcomes.

In addition, the NARDL model has recently been used for measuring the price transmission asymmetry in different countries [34–37]. In this context and to the best of the author's knowledge no work has been done on the onion price transmission asymmetric analysis by using the NARDL approach in Bangladesh. Thus, the symmetrical level of onion price transmission and associated market power of the market participants is yet to discover for policy development. In addition, by determining the extent of asymmetric price transmission, policymakers and market participants can have a better understanding of how changes in prices at one level of the onion market can affect prices at other levels and the market's overall performance. This information can inform pricing policy, supply chain management, and other approaches aimed at enhancing market efficiency and reducing the volatility of the onion market in Bangladesh.

## Data description and econometric methodology

### Data description

The secondary data used in this study were collected from the Department of Agricultural Marketing, Bangladesh [38]. Hence, there was no requirement to obtain ethical approval or respondent's consent for this study. However, the data were obtained legally and according to the authority's terms and conditions. The data include the monthly average wholesale and retail price of local and imported onions. The data was covered the period from January 2006 to December 2020. The data period was confined to this specific period due to the several

reasons. Firstly, some monthly value of wholesale and retail prices are missing before 2006. Hence, we have used 15 years' monthly data for empirical analysis because of unavailability. Another reason is that the price of onion is a serious concern in recent years. The wholesale and retail price of onion in Bangladesh is fluctuating over the ten years. Therefore, we used 15 years monthly data for our study. Since price transmission occurs most frequently at secondary stage of market in Bangladesh, this study focused on wholesale and retail prices of onion. Moreover, imported onion prices have significantly influenced the prices of local onion, so it is quintessential to take the price level of imported onion along with the local onion. The whole-sale price of local onion is represented by LWP, whereas the wholesale price of imported onion is IWP. Moreover, the retail price of local onion is represented by LRP as well as the imported retail price of onion is expressed as IRP. The units of the price measurement are symbolized as Bangladeshi Taka (BDT) per kg.

### Econometric methodology

The analysis of time series data entails a number of procedures, one of which is to ensure that the research variables are stationary and that none of them are integrated beyond the second order. As a result, the unit root characteristics of the variables employed in the study must be scrutinized. When a variable has more than one order of integration, it produces erroneous results. To verify the order integration of variables, we used Augmented Dickey–Fuller (ADF) [39] and Phillips–Perron (PP) [40] unit root tests prior employing the time series econometric model. Before testing the causality, it's imperative to consider the cointegrating properties of the variables. We used the Bounds cointegration test to see if there was any cointegration between variables.

To investigate the asymmetry price transmission between local and imported onion in Bangladesh, the study took NARDL approach for examining the relationship between onion prices. This method was developed by Shin et al. [41]. The NARDL model is used to analyze the asymmetric non-linearity relationship between the research variables; it also aids in determining the likelihood of asymmetric influence of explanatory variables' positive and negative shocks on the dependent variable in the long and short run [42]. In macroeconomic analysis, the short run illustrates a timeframe during which prices do not respond to alterations in overall economic conditions whereas the long run indicates such a period during which prices and some other economic variables (i.e., wages) become flexible [43].

As a result, the NARDL model was used in this research. The NARDL approach can be used when the series is stationary at level, at the first difference, and in mixed order of integration even though the sample is small [41]. From the findings of the causality test, this study has applied two linear model which are stated in the following equations:

$$\text{Model 1}: \text{LWP}_t = \alpha_0 + \beta_1 \text{IWP}_t + \mu_t \tag{1}$$

$$\text{Model 2}: LRP_t = \theta_0 + \lambda_1 IRP_t + \varepsilon_t \tag{2}$$

Where LWP is the local wholesale price of onion, LRP donates the local retail price of onion, IWP indicates the imported wholesale price of onion, IRP is the imported retail price of onion, t denotes the time period, and α, β, θ, and λ are the parameters to be estimated, whereas μ, and ε are the error correction term. The asymmetric cointegration equation of

above models are as follows:

$$\text{LWP}_t = \alpha_0 + \beta_1 \text{IWP}_t^+ + \beta_2 \text{IWP}_t^- + \mu_t \tag{3}$$

$$\text{LRP}_t = \theta_0 + \lambda_1 \text{IRP}_t^+ + \lambda_2 \text{IRP}_t^- + \varepsilon_t \tag{4}$$

where $\text{IWP}_t^+$, $\text{IWP}_t^-$, $\text{IRP}_t^+$, and $\text{IRP}_t^-$ indicates the partial sum of positive and negative changes in IWP, and IRP, respectively. The imported wholesale and retail prices will be decomposed into two new variables that will represent positive and negative shocks in imported wholesale and retail prices, as follows:

$$\text{IWP}_t^+ = \sum_{i=1}^{t} \Delta \text{IWP}_t^+ = \sum_{i=1}^{t} \max(\Delta \text{IWP}_i, 0)$$

$$\text{IWP}_t^- = \sum_{i=1}^{t} \Delta \text{IWP}_t^- = \sum_{i=1}^{t} \min(\Delta \text{IWP}_i, 0)$$

$$\text{IRP}_t^+ = \sum_{i=1}^{t} \Delta \text{IRP}_t^+ = \sum_{i=1}^{t} \max(\Delta \text{IRP}_i, 0)$$

$$\text{IRP}_t^- = \sum_{i=1}^{t} \Delta \text{IRP}_t^- = \sum_{i=1}^{t} \min(\Delta \text{IRP}_i, 0)$$

For calculating the asymmetric long-run and short-run relationship among study variables, we replace Eqs (3) and (4) into Eqs (5) and (6), respectively as described in Shin et al. (2014) [41]:

$$\Delta \text{LWP}_t = \alpha_0 + \alpha_1 \text{LWP}_{t-1} + \alpha_2 \text{IWP}_t^+ + \alpha_3 \text{IWP}_t^- + \sum_{i=1}^{p} \beta_i \Delta \text{LWP}_{t-1} + \sum_{i=1}^{q} \lambda_{2i} \Delta \text{IWP}_{t-1}^+ + \sum_{i=1}^{r} \lambda_{3i} \Delta \text{IWP}_{t-1^-} + \varphi \text{ECM}_{t-1} + \mu_t \tag{5}$$

$$\Delta \text{LRP}_t = v_0 + v_1 \text{LRP}_{t-1} + v_2 \text{IRP}_t^+ + v_3 \text{IRP}_t^- + \sum_{i=1}^{p} \omega i \Delta \text{LRP}_{t-1} + \sum_{i=1}^{p} \psi_{2i} \Delta \text{IRP}_{t-1}^{+} + \sum_{i=1}^{r} \psi_{3i} \Delta \text{IRP}_{t-1}^{-} + \eta \text{ECM}_{t-1} + \phi_t \tag{6}$$

where $\Delta$ first difference parameter in time period, p, q, r indicates the respective lag order, parameters with $\Sigma$ symbol denotes the short-run coefficient, and others are long-run coefficients, $\varphi$, and $\eta$ represents the elasticity of lag error correction term, $\mu$, and $\phi$ are the error correction term.

The existence of long-term cointegration among the variables in Eqs (5) and (6) is determined by performing a standard F-test. Precisely, the null hypothesis of no co-integration is tested. When the calculated F-statistics is higher than the upper bound critical value, the null hypothesis can be rejected [25]. If there is co-integration among variable found then the asymmetry price transmission of (5), and (6) can be discussed in the short-run and long-run. The estimated findings of Eqs (5) and (6) can be examined through the Wald test [44] to determine the possible asymmetric effect of independent variables on the dependent variable. After identifying the presence of long- and short-term asymmetric effects on research variables, we utilize a variety of diagnostic tests to evaluate the robustness of our findings. We employed the Breusch–Godfrey LM test for serial correlation [45], whereas, to check heteroscedasticity we used the Breusch–Pagan–Godfrey test, and for model stability we applied the Ramsey Reset test [46] CUSUM and CUSUM Square test.

According to Tiffin and Dawson (2000), the empirical estimation of price transmission asymmetry from the wholesale to retail level or vice versa depend on the causality relationships [47]. The Granger causality test [42] has been widely used with time series data to describe the

causality relationship between economic variables. This test uses past information for one variable to predict the other. The current study has investigated the causality relationship between wholesale and retail prices of onion to determine if the past and present information at one level of the marketing channel can be used to improve the forecast of the future prices at another level. The test is based on the following model variables respectively:

$$LWP_t = a_0 + \sum_{j=1}^{k} \alpha_j LWP_{t-j} + \sum_{j=1}^{k} \beta_{t-j} IWP_{t-j} \tag{7}$$

$$LRP_t = a_0 + \sum_{j=1}^{k} \alpha_j LRP_{t-j} + \sum_{j=1}^{k} \beta_{t-j} IRP_{t-j} \tag{8}$$

The null hypothesis of the Granger test is that no Granger causality exists between the variables based on the test. Therefore, the rejection of the null hypothesis indicates that the causality relationship refers to unidirectional causality, bidirectional causality, or independence.

## Results and discussion

### Different price level in the onion supply chain

Fig 1 shows the trends of three different price level in the supply chain of Local onion in Bangladesh for the period of January 2006 to December 2020. It has been shown that there is a great variability in the prices of onion. Moreover, all the prices indicate a general trend which implies a co-integration relationship between the variables. From the trend of farm price, the highest price was observed at 142 Tk/Kg in November 2019 whereas the least price was 14 BDT/Kg in February 2019. Likewise, both the wholesale and retail prices of onion were also peaked in November 2019 (i.e., 158 BDT/Kg and 168 BDT/Kg) whereas the wholesale and retail prices were also least in February 2019 (i.e., 16 BDT/Kg and 20 BDT/Kg, respectively).

### Descriptive statistics and unit root test

Before analyzing the results, it is required to depict the descriptive statistics for the variable that was utilized in the study. Table 1 presents the findings of descriptive statistics. The means of the variables are nearly similar to their medians, and they are all positively skewed. According to the kurtosis values, none of the variables are mesokurtic, while all of the variables are leptokurtic. The Jarque-Bera test [48] statistic and probability unveil that the residuals of all variable are not distributed normally which allow us to employ the NARDL model precisely. The findings of the unit root test, namely Augmented Dickey–Fuller (ADF) test and The Phillips–Perron (PP) test, are shown in Table 2. Before estimating the elasticities, it is required to analyze the order of integration among variables using the appropriate unit root test. Since the cointegration approach and causality assessment rely on variable integration, hence it is pertinent to affirm the stationary properties. The findings of the ADF test [39] show that local wholesale and retail onion prices are stationary at both the level and first difference. At the first difference, however, all of the variables become stationary. As a result, the ADF test confirms a common order of integration. However, we employed another stationary test. The Phillips–Perron (PP) [40] test is used to adjust the t test statistics non-parametrically. The non-specific autocorrelation and heteroscedasticity in the disturbance process of the test equation have no effect on this test. The results of the PP test reveal that at the first difference level, all variables are stationary. Importantly, none of the variables have been determined to be stationary at the second difference, allowing us to proceed with the NARDL model.

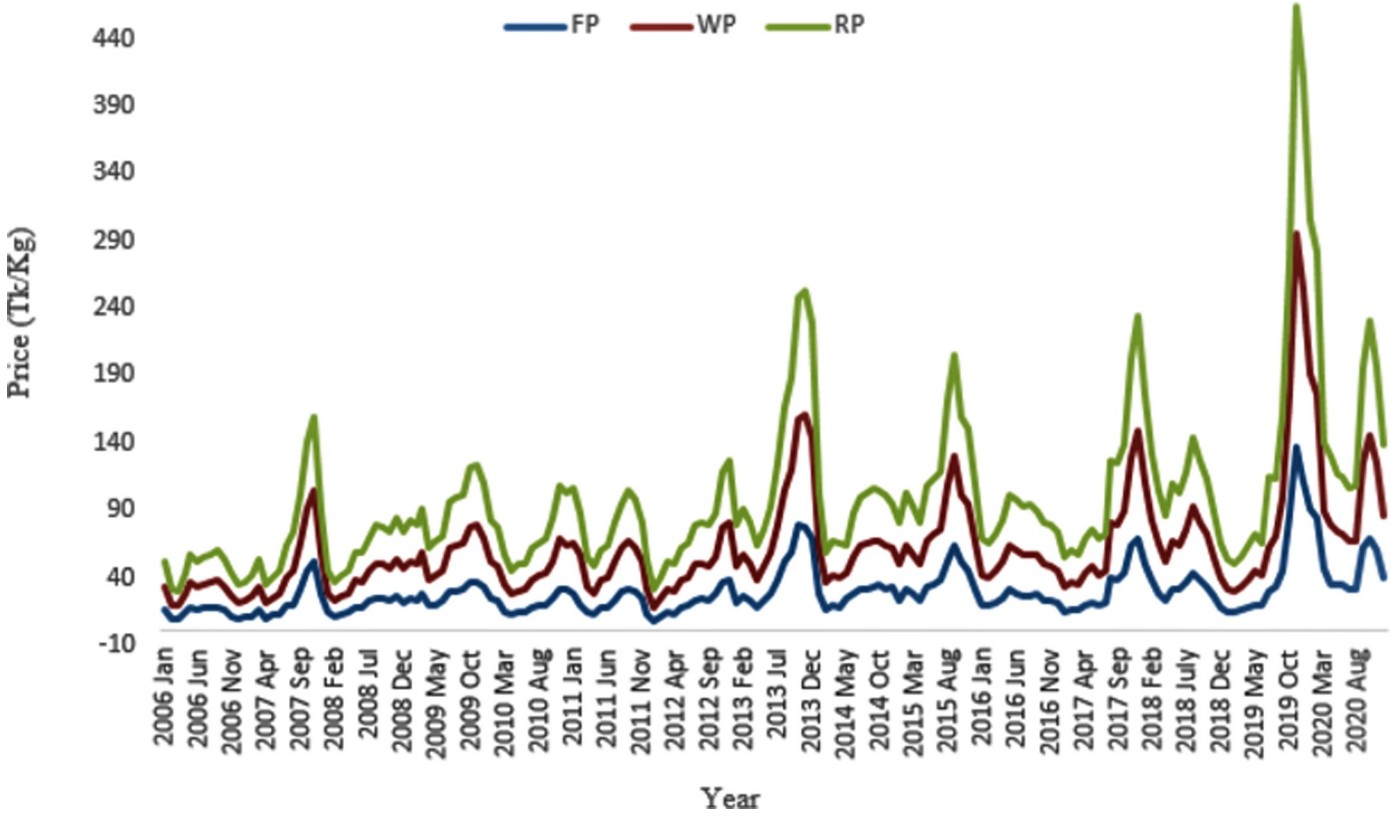

**Fig 1. The trend of different onion price level.** FP = Farm Price, WP = Wholesale Price and RP = Retail Price of Onion in Bangladesh.

## F bound test

After checking the integration order, it's time to look at the cointegrating attributes of the variables. We used the Bounds cointegration tests (F statistics and t statistics) to see if there was any cointegration between variables. The test methods depend on the upper and lower critical bounds. If the empirical values of F and t statistics exceed the upper bound, the null is rejected (i.e. the variables are cointegrated). Moreover, If the values lie below the lower bound, the variables are not cointegrated and if they lie between the critical bounds, the test is inconclusive. From the empirical results show that both the F statistics and t statistics value is greater than

**Table 1. Descriptive statistics of wholesale and retail price of onion (local and imported).**

|  | Local wholesale price | Local retail price | Imported wholesale price | Imported retail price |
|---|---|---|---|---|
| Mean | 3.348783 | 3.476812 | 3.183942 | 3.330837 |
| Median | 3.303217 | 3.421484 | 3.087397 | 3.258097 |
| Maximum | 5.063101 | 5.123964 | 4.956601 | 5.049856 |
| Minimum | 2.299581 | 2.397895 | 2.400619 | 2.564949 |
| Std. Dev. | 0.496963 | 0.481008 | 0.478344 | 0.464504 |
| Skewness | 0.599528 | 0.615901 | 0.997153 | 0.969086 |
| Kurtosis | 3.732941 | 3.814508 | 3.921481 | 4.095328 |
| Sum | 602.7810 | 625.8262 | 573.1096 | 599.5506 |
| Sum Sq. Dev. | 44.20806 | 41.41499 | 40.95748 | 38.62168 |
| Observations | 180 | 180 | 180 | 180 |

**Table 2. Unit root test (ADF and PP test).**

| Variable | ADF test | | PP test | |
|---|---|---|---|---|
| | At level | At first difference | At level | At first difference |
| Local wholesale price | -2.657924*** | -9.432093*** | -1.478602 | -9.453778*** |
| Local retail price | -2.412706** | -9.650550*** | -1.586269 | -9.490182*** |
| Imported wholesale price | -0.499557 | -9.098001*** | -0.341093 | -8.586239*** |
| Imported retail price | 0.071751 | -8.545336*** | -0.246055 | -8.511746*** |

ADF = Augmented Dickey–Fuller test and PP = The Phillips–Perron test

Note: Both

*** and

** Show significance at 1% and 5% level respectively

the upper bound value, indicating a long-term relationship between variables (Table 3). The results of both the tests confirm that the variables in all three models are cointegrated, and for market asymmetry determination the variables should be cointegrated as suggested in prior research [7–9, 11, 24, 26]. In addition, NARDL F-statistic [49] confirms that all the variables have long-run asymmetric association in the onion supply chain of Bangladesh.

## Wald test

In accordance with the results of the Wald test [50] shown in Table 4, this study investigates the long run and short run symmetry of the selected variables using two distinct models. For model 1, the test indicates that the long run asymmetry is not significant, however the short run asymmetry is. This shows that the null hypothesis is accepted over the long run but rejected over the short run. The results indicate that when the cost of onions rises, the wholesale price of local onions may increase more quickly than the wholesale price of imported onions. In contrast, the Wald test result for model 2 reveals that the null of long run symmetry is rejected strongly, whilst the null of short run symmetry is accepted. In the long run, the study suggests that the retail price of imported onion is not transmitted equally to the retail price of local onion, although in the short run, the transmission is equal. No prior similar study between import and local product market was found but the most relevant study revealed equal price transmission is possible between the wholesale and retail market [26] which align with this study result. The findings of the study imply that policymakers in Bangladesh must take into account the diverse dynamics of market adjustment when establishing policies to manage onion prices.

## Empirical NARDL cointegration model

According to the results reported in Table 5, the empirical analysis of two models generated by Eqs (5) and (6) indicates that all estimated parameters are significant at the 1%. The calculated long-run coefficient for the wholesale price of imported onion in model 1 shows that positive shocks have a 1.090% influence, while negative shocks have a 1.394% effect on the wholesale price of local onion. This indicates that a 1% increase (decrease) in the wholesale price of imported onion would result in a 1.090% (1.394%) increase (decrease) in the wholesale price of local onion. These results indicate that negative shocks in the wholesale prices of imported onion have a higher effect on local wholesale prices than positive ones, showing that imported onion prices are more responsive to a fall in the wholesale price of local onion. In the short run, the findings suggest that the wholesale price of imported onion and the wholesale price of local onion have a symmetric relationship. This indicates that both positive and negative

**Table 3. Bound cointegration test.**

| F-statistics | Level of significance | Lower bound I(0) | Upper bound I(1) | Long-run relationship |
|---|---|---|---|---|
| | | **Model 1** | | |
| 15.39 | 10% | 4.19 | 5.06 | Present |
| | 5% | 4.87 | 5.85 | |
| | 1% | 6.34 | 7.52 | |
| | | **Model 2** | | |
| 13.44 | 10% | 2.63 | 3.35 | Present |
| | 5% | 3.1 | 3.87 | |
| | 1% | 4.13 | 5.00 | |
| **t-statistics** | **Level of significance** | **Lower bound I(0)** | **Upper bound I(1)** | **Long-run relationship** |
| | | **Model 1** | | |
| -5.882588 | 10% | -3.13 | -3.63 | Present |
| | 5% | -3.41 | -3.95 | |
| | 1% | -3.96 | -4.53 | |
| | | **Model 2** | | |
| -4.376514 | 10% | -3.13 | -3.63 | Present |
| | 5% | -3.41 | -3.95 | |
| | 1% | -3.96 | -4.53 | |

shocks to the wholesale price of imported onion have a substantial negative effect on the wholesale price of local onion. Specifically, positive shocks in the price of imported onion in the current year, as well as in lags 5 and 6, have a negative impact on the wholesale price of local onion. This means that an increase in the price of imported onion results in a fall in the price of locally produced onion. On the other hand, negative shocks in the imported onion price have a large negative impact on the wholesale price of local onion at different lags. This means that a fall in the price of imported onion results in a rise in the price of local onion. The asymmetric price transmission was also observed by Mgale [10], Rajendran [24], and Kumar [26] in the both long and short run in Tanzanian and Indian onion market, respectively. Overall, the link between the wholesale price of imported onion and the wholesale price of local onion shows that policies focused at stabilizing the price of imported onion could also contribute to stabilizing the price of local onion. It also means that the local onion market is susceptible to foreign influences and can be influenced by fluctuations in the global onion market.

In case of model 2, the estimated long-run coefficients for both positive and negative shocks to the imported retail price of onion have a significant effect on the local retail price of onion.

**Table 4. Testing the presence of asymmetries using Wald test.**

| Variable | F-statistic | P-value | Presence of asymmetry |
|---|---|---|---|
| | **Model 1** | | |
| IWP (Long-run) | 1.710582 | 0.1928 | No |
| ΔIWP (Short-run) | 2.827599** | 0.0546 | Yes |
| | **Model 2** | | |
| IRP (Long-run) | 9.276738** | 0.0027 | Yes |
| ΔIRP (Short-run) | 0.357377 | 0.5508 | No |

Note:

** and

* Show significance at 5% and 10% level respectively

**Table 5. NARDL estimation results.**

| Model 1: Imported wholesale to local wholesale price transmission | | | |
|---|---|---|---|
| Variables | Coefficient | Standard error | t-statistics |
| C | 0.895295*** | 0.154071 | 5.810946 |
| $\ln LWP_{(t-1)}$ | -0.307177*** | 0.052218 | -5.882588 |
| $\ln IWP^+_{(t-1)}$ | 0.334874*** | 0.060530 | 5.532349 |
| $\ln IWP^-_{(t-1)}$ | 0.428315*** | 0.071851 | 5.961193 |
| $\Delta \ln IWP^+$ | -0.717097*** | 0.064703 | -11.08282 |
| $\Delta \ln IWP^+_{(t-1)}$ | 0.049999 | 0.079035 | 0.632623 |
| $\Delta \ln IWP^+_{(t-2)}$ | 0.029195 | 0.078915 | 0.369958 |
| $\Delta \ln IWP^+_{(t-3)}$ | 0.039871 | 0.079734 | 0.500053 |
| $\Delta \ln IWP^+_{(t-4)}$ | -0.114640 | 0.087218 | -1.314415 |
| $\Delta \ln IWP^+_{(t-5)}$ | -0.267762** | 0.084708 | -3.161009 |
| $\Delta \ln IWP^+_{(t-6)}$ | -0.235547** | 0.083327 | -2.826785 |
| $\Delta \ln IWP^-$ | 0.955522*** | 0.077254 | 12.36855 |
| $\Delta \ln IWP^-_{(t-1)}$ | -0.116936 | 0.079759 | -1.466117 |
| $\Delta \ln IWP^-_{(t-2)}$ | -0.162323** | 0.073846 | -2.198120 |
| $\Delta \ln IWP^-_{(t-3)}$ | -0.215948** | 0.071384 | -3.025176 |
| $\Delta \ln IWP^-_{(t-4)}$ | -0.074925 | 0.070341 | -1.065170 |
| $\Delta \ln IWP^-_{(t-5)}$ | -0.195055** | 0.069994 | -2.786752 |
| $R^2$ | 0.835211 | | |
| Adjusted $R^2$ | 0.819366 | | |
| Durbin-Watson statistic | 1.817889 | | |
| Asymmetry statistics: | Long-run effect [+] | | Long-run effect [–] | |
| Exogenous variables | Coefficient | Standard error | Coefficient | Standard Error |
| lnIWP | 1.0901*** | 0.1577 | 1.3943*** | 0.2281 |

| Model 2: Imported retail to local retail price transmission | | | |
|---|---|---|---|
| Variables | Coefficient | Standard error | t-statistics |
| C | 1.379373*** | 0.189762 | 7.268977 |
| $\ln LRP_{(t-1)}$ | -0.454611*** | 0.062809 | -7.237948 |
| $\ln IRP^+_{(t-1)}$ | 0.391268*** | 0.060258 | 6.493165 |
| $\ln IRP^-_{(t-1)}$ | 0.381868*** | 0.059835 | 6.382054 |
| $\Delta \ln LRP_{(t-1)}$ | 0.117910*** | 0.044849 | 2.629052 |
| $\Delta \ln IRP^+$ | 0.794813*** | 0.074999 | 10.59762 |
| $\Delta \ln IRP^-$ | 0.872109*** | 0.083776 | 10.41003 |
| $R^2$ | 0.942337 | | |
| Adjusted $R^2$ | 0.940314 | | |
| Durbin-Watson statistic | 2.120410 | | |
| Asymmetry statistics: | Long-run effect [+] | | Long-run effect [–] | |
| Exogenous variables | Coefficient | Standard error | Coefficient | Standard Error |
| lnIRP | 0.8606*** | 0.0565 | 0.8399*** | 0.0599 |

NARDL = Non-linear Auto Regressive Distributed Lag Model

Note: Both

*** and

** Show significance at 1% and 5% level respectively

The fact that the coefficient for positive shocks ($\ln IRP^+$ = 0.8606) is greater than that for negative shocks ($\ln IRP^-$ = 0.8399), indicating that the transmission of positive shocks is more intense. This shows that the retail price of imported onion reacts more quickly to an increase

in the retail price of local onion. Based on the findings, it is feasible to conclude that the onion supply chain in Bangladesh possesses some degree of market power. The positive and statistically significant effect of imported onion prices on local onion retail prices implies that onion sellers, including both domestic producers and importers, may have some influence on market prices. Positive shocks in imported onion prices had a stronger impact on local onion prices than negative shocks, which may suggest that onion traders have some power to increase prices when demand is high. This may be the result of limited competition among sellers or the presence of entry obstacles that make it difficult for new sellers to enter the market and increase competition.

However, the findings indicate an asymmetric relationship between the local onion retail price and the imported onion retail price in the short run. Both positive and negative shocks to the local onion retail price have a considerable positive effect on the retail price of imported onions with varying lags. Yet, the impact of positive and negative shocks to the retail price of imported onion on the local onion retail price are distinct. A positive shock in the retail price of imported onion in the current year and in lag 1 has a positive influence on the retail price of local onion, meaning an increase in the price of imported onion results in a drop in the price of local onion. A negative shock in the retail price of imported onion in the current year has a positive influence on the retail price of local onion, indicating that a fall in the price of imported onion results in a rise in the price of local onion. The short run asymmetric transmission may have been the result of adjustment costs, such as the storage cost of onion. This asymmetric relationship between the local onion retail price and the imported onion retail price shows that policies designed to stabilize the local onion price could also have a positive impact on the imported onion price. It also implies that the local onion market may have some control over the price of imported onions, but that the relationship is complex and subject to other influences.

The dynamic multipliers assist us in determining the evolution of a price at a specific level of the supply chain in response to a shock to a price at a different level of the chain, so offering a picture of the path to the new equilibrium. The dynamic multipliers for price transmission from the local wholesale to the imported wholesale price of onion are shown in Fig 2 and the dynamic multipliers for price transmission from local retail to imported retail price of onion are depicted in Fig 3. The X-axis represents the time period, and Y-axis represents the magnitude of response of local wholesale onion price and imported retail price while positive and negative shocks have been given in imported wholesale onion price in Fig 2 and imported retail in Fig 3, respectively. We see a response in onion wholesale prices in the local market in case of both positive and negative shock in the partial sum of imported price of onion from other countries. But local retail price of onion responds more to negative shock in the partial sum of imported local onion price than its positive counterpart, as shown in Fig 3.

Table 6 displays the findings of the diagnostic tests. The p values for Ramsey Reset test [46] was 0.2430 for model 1 and 0.2531 for model 2 showing that both models have no specification errors. It reveals that the models are correctly specified. The LM and Breusch-Pagan-Godfrey tests' [45, 51] respective p values for model 1 were 0.1707 and 0.1227, whereas they were 0.2636 and 0.5996 for model 2, indicating that the model is devoid of serious serial correlation problems and heteroscedasticity.

To guarantee parameter stability in the model, the robustness of each statistical research must be examined. The parameter stability tests such as CUSUM and CUSUMSQ were applied as a consequence. We may examine the model's stability after calculating the long-run and short-run. The predicted parameter is stable if the CUSUM and CUSUMSQ lines remain inside the top and lower boundaries of the CUSUM and CUSUMSQ graph. The blue line of

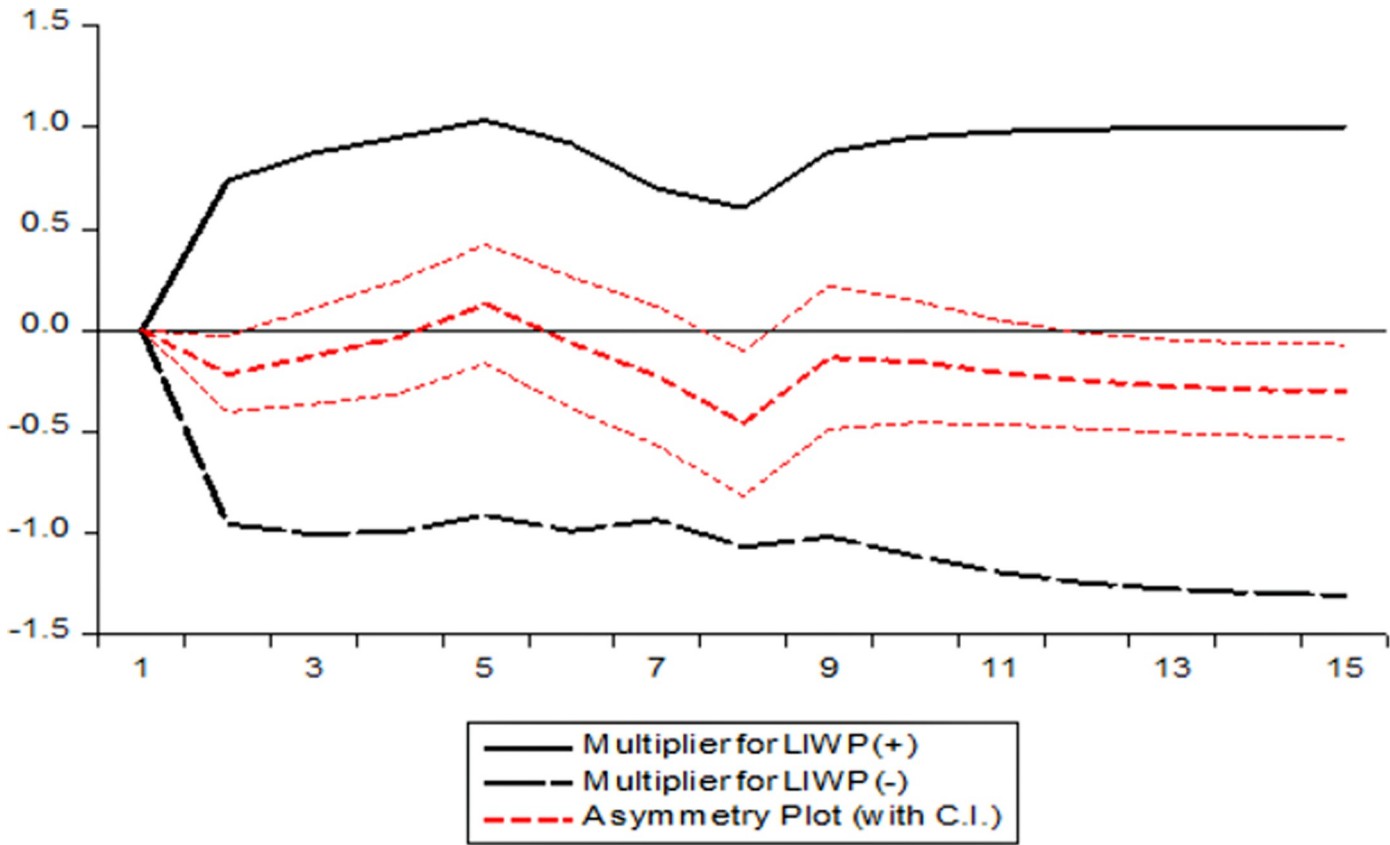

**Fig 2. Dynamic multiplier graph for model 1.** LIWP⁺ = Log of Imported Wholesale Price (Positive), LIWP⁻ = Log of Imported Wholesale Price (Negative).

the upper and lower bound stays inside the red line of the CUSUM and CUSUM Square graphs, demonstrating the stability of both models (Figs 4 and 5).

For empirical analysis of price transmission asymmetry, it is pertinent to select the causal markets. Here, for identifying the causal markets the study applied Granger Causality test [42]. The test has been applied to two pairs of price levles, including local wholesale price and imported wholesale price (LWP, IWP) and local retail price and imported retail price (LRP, IRP). Table 7 reveals that, for pairs of LWP and IWP with a significance level of 5% or less, there is a unidirectional causal relationship between the imported wholesale price and the local wholesale price of onion. Moreover, for the pair LRP and IRP and at all reasonable level of significance, causality also shows unidirectionally from imported retail price to local retail price of onion in Bangladesh. Given that 40% of domestic demand is met by imported onions from neighboring countries, these unidirectional causal relationships further clarify the impact of imported onions on retail onion prices for consumers to purchase.

## Limitations

This study only looked at the Bangladeshi onion market, but it might also apply to other commodities there and in other nations. It is worth noting that these findings may be influenced by the specific characteristics of the onion market in Bangladesh, and may not necessarily apply to other countries or other agricultural markets. Further research is needed to confirm these findings and to explore the reasons behind the observed differences in the transmission of shocks between positive and negative shocks in the onion market. Moreover, this study

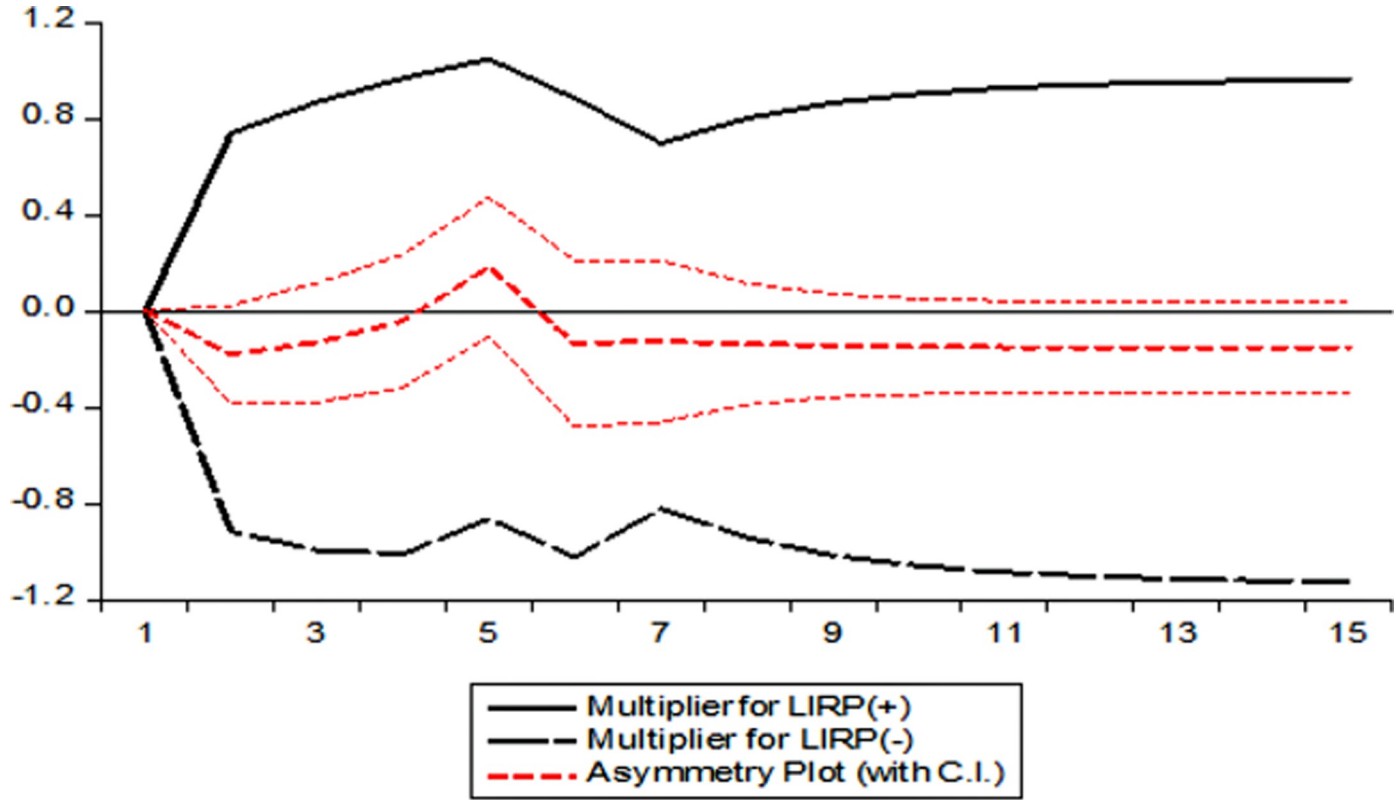

**Fig 3. Dynamic multiplier graph for model 2.** $LIRP^+$ = Log of Imported Retail Price (Positive), $LIRP^-$ = Log of Imported Retail Price (Negative).

ignores the inclusion of additional variables in the model and solely takes into account price transmission. Therefore, the particularity of onion production capacity, soil suitability, and policy intervention variables could be taken into consideration in future research since some characteristics related to the particularity of the area, such as the requirements of LFA (less favored area) and non-LFA areas, where the production cost differs significantly. Additional research might be conducted while taking the agriculture policy factors into account.

**Table 6. Diagnostic tests.**

| Test | F- Statistic | Probability |
|------|-------------|-------------|
| **Model 1** | | |
| Ramsey RESET Test | 1.146577 | 0.2430 |
| LM for serial correlation | 1.690885 | 0.1707 |
| BPG for heteroscedasticity | 1.704353 | 0.1227 |
| CUSUM | Stable | |
| CUSUM square | Stable | |
| **Model 2** | | |
| Ramsey RESET Test | 1.314640 | 0.2531 |
| LM for serial correlation | 1.343807 | 0.2636 |
| BPG for heteroscedasticity | 0.763422 | 0.5996 |
| CUSUM | Stable | |
| CUSUM square | Stable | |

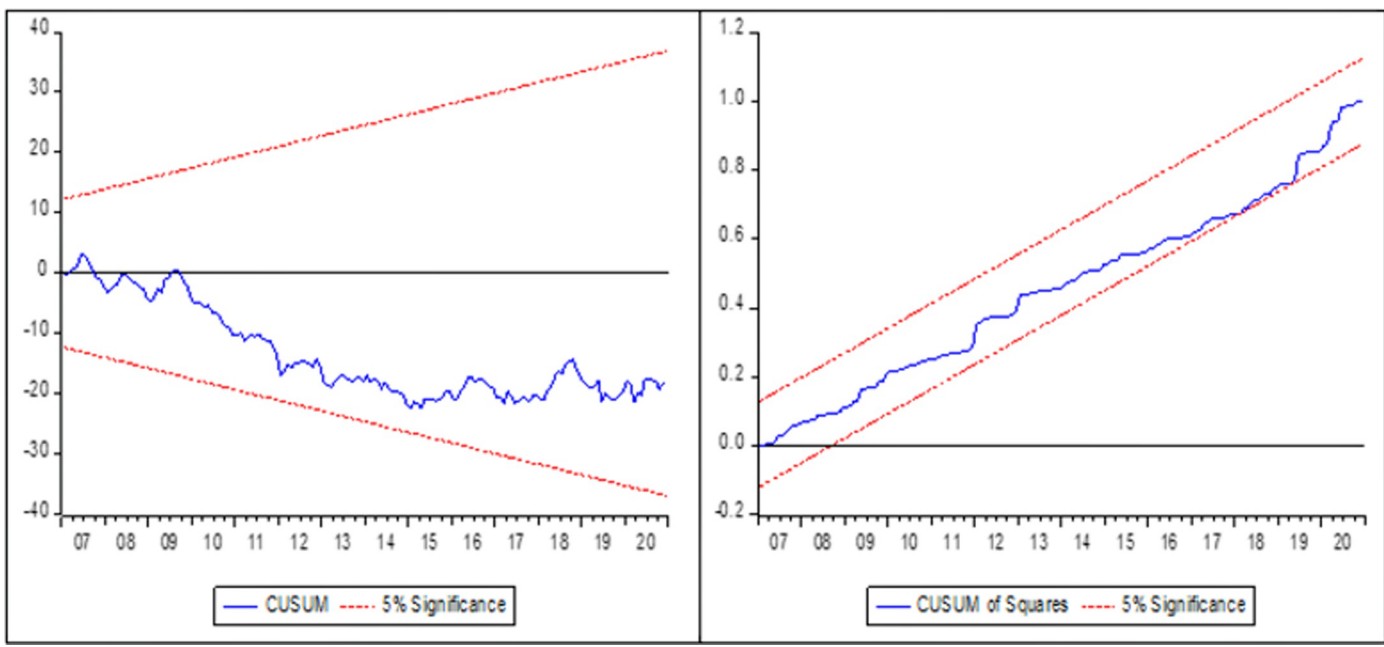

**Fig 4. CUSUM and CUSUM square test for model 1.**

## Concluding Remarks

The study examined the long-run and short-run symmetry between the wholesale and retail prices of imported and local onions in Bangladesh using two distinct models. The aim of the study was to assess the responses of local onion prices to a positive or negative change in the price of imported onions at the wholesale and retail levels. The results of the Wald test show

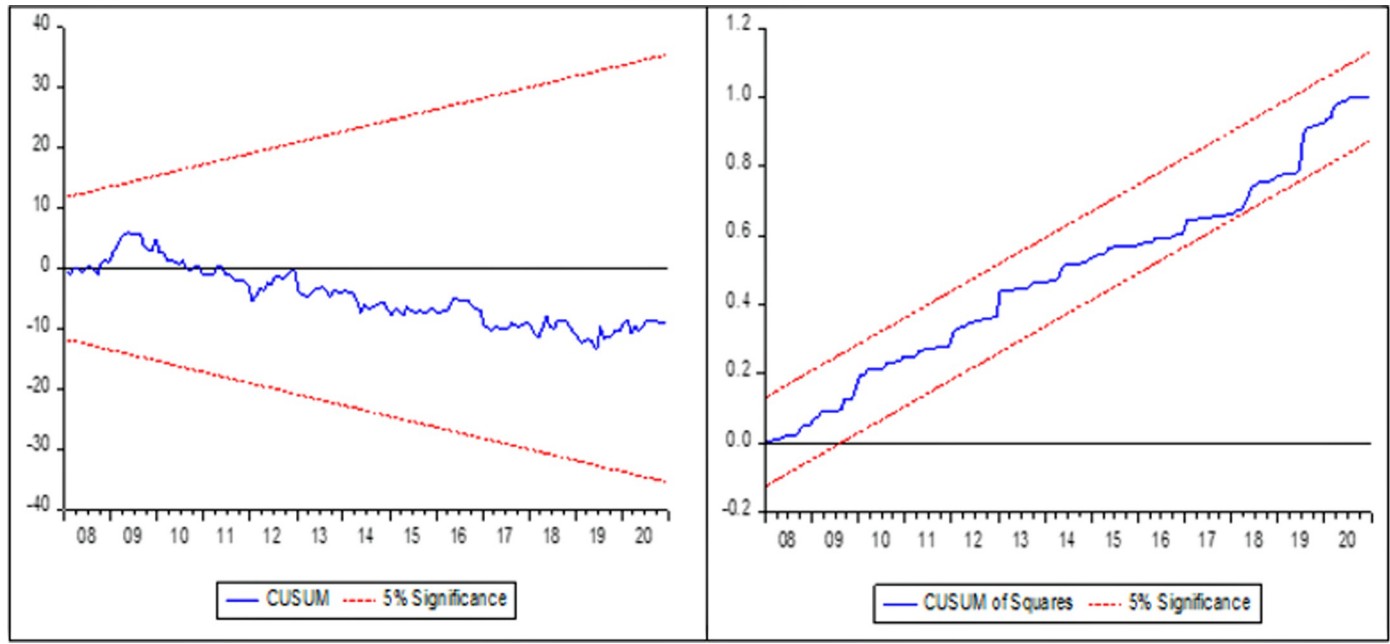

**Fig 5. CUSUM and CUSUM square test for model 2.**

**Table 7. Pair-wise Granger Causality test.**

| Null Hypothesis ($H_0$) | F statistics | Probability |
|---|---|---|
| IWP does not Granger Cause LWP | 3.26908 | 0.0130** |
| LWP does not Granger Cause IWP | 1.69584 | 0.1533 |
| IRP does not Granger Cause LRP | 5.20475 | 0.0006** |
| LRP does not Granger Cause IRP | 1.36847 | 0.2470 |

Note: Both

*** and

** Show significance at 1% and 5% level respectively

that the long-run asymmetry is not significant in model 1, but the short-run asymmetry is significant. In model 2, the null hypothesis of long-run symmetry is strongly rejected, while the null of short-run symmetry is accepted. The findings of the NARDL cointegration models suggest that when the cost of onions rises, the wholesale price of local onions may increase more quickly than the wholesale price of imported onions in the short run, while the retail price of imported onions is not transmitted equally to the retail price of local onions in the long run. The study also reveals an asymmetric relationship between the retail prices of imported and local onions in the short run, while the relationship is symmetric in the long run. The findings also suggest the existence of a short-run adjustment cost and a long-run market power that affect the price transmission of onion at wholesale and retail levels. On the basis of the findings of the study, it would appear that there is a substantial opportunity to boost onion yield in Bangladesh to stabilize the domestic market price of onion.

Based on the results of the empirical analysis of price transmission asymmetry between local and imported onions in Bangladesh, policymakers should take into account the diverse dynamics of market adjustment when establishing policies to manage onion prices. It is recommended that policymakers should consider the development of policies to promote competition in the onion supply chain to prevent market power and ensure that onion prices are kept at reasonable levels. In addition, policies should be developed to support the storage and distribution of onions to prevent short-run asymmetries and ensure stable prices. It may also be useful to explore alternative methods of managing onion prices, such as public-private partnerships that involve the government and private sector working together to manage onion prices effectively. Overall, the study highlights the importance of considering the dynamics of price transmission asymmetry and market power when developing policies to manage onion prices in Bangladesh.

## Supporting information

**S1 Data. Raw data.**
(XLS)

## Acknowledgments

The authors would like to express their gratitude to the Research Management Committee (RMC) of Bangabandhu Sheikh Mujibur Rahman Agricultural University and the Department of Agricultural Marketing (DAM) for their assistance in research and data collection process, which made this study feasible. The authors gratefully acknowledged Plos One editor and the anonymous reviewers for their comments, insightful suggestions and careful reading of the manuscript.

## Author Contributions

**Conceptualization:** Farhana Arefeen Mila, Ashrafun Nahar.

**Data curation:** Farhana Arefeen Mila, Ashrafun Nahar.

**Formal analysis:** Farhana Arefeen Mila, Ashrafun Nahar, Md. Emran Hossain,
Md. Ruhul Amin.

**Funding acquisition:** Farhana Arefeen Mila, Ashrafun Nahar.

**Investigation:** Farhana Arefeen Mila, Ashrafun Nahar.

**Methodology:** Farhana Arefeen Mila, Ashrafun Nahar, Md. Emran Hossain,
Md. Ruhul Amin.

**Project administration:** Farhana Arefeen Mila, Ashrafun Nahar.

**Resources:** Farhana Arefeen Mila, Ashrafun Nahar.

**Software:** Farhana Arefeen Mila, Ashrafun Nahar.

**Supervision:** Farhana Arefeen Mila, Ashrafun Nahar.

**Validation:** Farhana Arefeen Mila, Ashrafun Nahar.

**Visualization:** Farhana Arefeen Mila, Ashrafun Nahar.

**Writing – original draft:** Farhana Arefeen Mila, Ashrafun Nahar, Md. Emran Hossain,
Md. Ruhul Amin.

**Writing – review & editing:** Farhana Arefeen Mila, Ashrafun Nahar, Md. Emran Hossain,
Md. Ruhul Amin.

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
