## [Editor Report · Decision Letter 0]

6 Nov 2022

PONE-D-22-26560Spatial Price Transmission in the Onion Markets of Bangladesh: An Application of NARDL ApproachPLOS ONE

Dear Dr. Nahar¶,

Thank you for submitting your manuscript to PLOS ONE. After careful consideration, we feel that it has merit but does not fully meet PLOS ONE’s publication criteria as it currently stands. Therefore, we invite you to submit a revised version of the manuscript that addresses the points raised during the review process.

We look forward to receiving your revised manuscript.

Kind regards,

Noman Arshed, PhD

Academic Editor

PLOS ONE

Journal Requirements:

"The authors would like to thank the Research, support and publication division of the University Grants Commission, Bangladesh (S.R. No: 6(79)/B.Mo.ko/B.O.Pro/Crop-16/2018/7032) for arranging fund to carry out this research work."

"No potential conﬂict of interest was reported by the author(s)."

Additional Editor Comments:

The gap discussed in the study is weak. author need to provide statistics about the proportion of imported onions in the total demand for onions so that this study can be deemed feasible.

Further if the resources used to produce local onions are same as imported onions they definitely their market prices will move together.

What about general cost of production in the country and the exchange rate effect, which will create the difference between domestic onions and imported onions.

Model may have shown high coefficients because of missing variable bias.

CUSUM of squares show that model is not stable it is touching the boundaries.

Authors have not ensured that the data is normally distributed so how come the inference can be applied let alone the asymmetric effects.

Since the model is not stable, authors cannot use the outcomes for policy implications.
---

## [Author Response · Author response to Decision Letter 0]

21 Dec 2022

Dear Editors,

Greetings of the day. Thank you very much for your kind efforts and providing the valuable comments to improve our manuscript. We have addressed all the comments as per the reviewers' instruction. We hope that the changes made in the revised version will be acceptable.

Please let me know If you need any other necessary documents or corrections. 

I am looking forward to hearing from you soon.

Thank you once again.

Best Regards

Ashrafun Nahar

---

## [Decision Letter · Decision Letter 1]

14 Feb 2023

PONE-D-22-26560R1Spatial Price Transmission in the Onion Markets of Bangladesh: An Application of NARDL ApproachPLOS ONE

Dear Dr. Nahar¶,

Thank you for submitting your manuscript to PLOS ONE. After careful consideration, we feel that it has merit but does not fully meet PLOS ONE’s publication criteria as it currently stands. Therefore, we invite you to submit a revised version of the manuscript that addresses the points raised during the review process.

We look forward to receiving your revised manuscript.

Kind regards,

Noman Arshed, PhD

Academic Editor

PLOS ONE

Reviewers' comments:

Reviewer's Responses to Questions

**Comments to the Author**

1. If the authors have adequately addressed your comments raised in a previous round of review and you feel that this manuscript is now acceptable for publication, you may indicate that here to bypass the “Comments to the Author” section, enter your conflict of interest statement in the “Confidential to Editor” section, and submit your "Accept" recommendation.

Reviewer #1: All comments have been addressed

Reviewer #2: (No Response)

2. Is the manuscript technically sound, and do the data support the conclusions?

Reviewer #1: Partly

Reviewer #2: Yes

3. Has the statistical analysis been performed appropriately and rigorously? 

Reviewer #1: Yes

Reviewer #2: Yes

4. Have the authors made all data underlying the findings in their manuscript fully available?

Reviewer #1: No

Reviewer #2: Yes

5. Is the manuscript presented in an intelligible fashion and written in standard English?

Reviewer #1: Yes

Reviewer #2: Yes

6. Review Comments to the Author

Reviewer #1: Spatial Price Transmission in the Onion Markets of Bangladesh: An Application of NARDL Approach

The paper is well written. Based on the previous comments, it seems to me that the authors have addressed them squarely. However, I have the following comments to strengthen the work:

1. The motivation for the study needs to be further strengthened. What is the reason for selection January 2006 to December 2020 for this study?

2. There is no literature review section for this study. Authors are advised to briefly review the related literature concerning the topic so that the gap the study will fill could be clear to the reader.

3. Since the emphasis is place on NARDL (see the topic) rather than causality, I suggest that the equations concerning NARDL should come first before causality. Moreover, causal relationship equations are not numbered. The equation should be numbered so that it will be easier to refer to it.

4. Why did the authors use bivariate model? There are other vital factors that can affect the interested variables. For robustness purposes, include other control variables.

5. Discuss clearly your major findings and compare the results with previously existing studies.

Reviewer #2: 1. The paper needs to clarify what is the meaning of investigating the asymmetric price transmission between the wholesale and retail markets for local and imported onion. Is the objective to identify price leaderships?

2. The authors should explain the definitions for short-run and long-run in the study.

3. Some discussions should be done on why the results might be difference between positive and negative shocks.

4. ln265 "due to the market power structure of the onion supply chain..." This part should be explained more in detail so that people not familiar with he Bangladesh onion market can understand.

5. ln341 "stabilize the domestic market price of onion..." Please explain why finding asymmetric adjustment is related to stabilizing the onion market. I am not sure if asymmetric pricing always means instability of markets. I also feel most agricultural price are not stable so the policy implication of the study is a little naive. Implications should be more directly connected to the finding of the study.

7. PLOS authors have the option to publish the peer review history of their article (what does this mean?). If published, this will include your full peer review and any attached files.

Reviewer #1: **Yes: **Ojonugwa Usman

Reviewer #2: No

---

## [Author Response · Author response to Decision Letter 1]

31 Mar 2023

Dear Reviewers and editors,

Thank you for your valuable comments and suggestions. We really appreciate all the efforts and insightful advices by the reviewers and editors to improve the research article. We have diligently reworked each section of the article based on the feedback. We have included all the tables in the manuscript. Additionally, we have uploaded the figure in the attached files as per the requirements. We believe that the modifications made to the new version will be acceptable.

Please kindly let me know if any further corrections or necessary documents needed. I would also like to let you know that we have uploaded the response of all the reviewers in the attached file. 

I am looking forward for your kind response. 

Thank you for your kind consideration. 

Sincerely,

Ashrafun Nahar

---

## [Editor Report · Decision Letter 2]

4 Apr 2023

Spatial Price Transmission in the Onion Markets of Bangladesh: An Application of NARDL Approach

PONE-D-22-26560R2

Dear Dr. Nahar¶,

We’re pleased to inform you that your manuscript has been judged scientifically suitable for publication and will be formally accepted for publication once it meets all outstanding technical requirements.

Kind regards,

Noman Arshed, PhD

Academic Editor

PLOS ONE
---

## [Editor Report · Acceptance letter]

10 Apr 2023

PONE-D-22-26560R2 

Spatial Price Transmission in the Onion Markets of Bangladesh: An Application of NARDL Approach 

Dear Dr. Nahar¶:

I'm pleased to inform you that your manuscript has been deemed suitable for publication in PLOS ONE. Congratulations! Your manuscript is now with our production department. 

Kind regards, 

on behalf of

Dr. Noman Arshed 

Academic Editor

PLOS ONE